# Galectins Are Central Mediators of Immune Escape in Pancreatic Ductal Adenocarcinoma

**DOI:** 10.3390/cancers14225475

**Published:** 2022-11-08

**Authors:** Zhengting Jiang, Wenjie Zhang, Gengyu Sha, Daorong Wang, Dong Tang

**Affiliations:** 1Clinical Medical College, Yangzhou University, Yangzhou 225000, China; 2Department of General Surgery, Institute of General Surgery, Clinical Medical College, Yangzhou University, Northern Jiangsu People’s Hospital, Yangzhou 225000, China

**Keywords:** galectins, galectin-1, PDAC, immune escape, immune cells, fibrosis, cellular metabolism

## Abstract

**Simple Summary:**

Pancreatic ductal adenocarcinoma (PDAC) is one of the most lethal cancers with a high degree of immune tolerance. Galectins induce induction of immune evasion behavior in tumor cells. Galectins each play a role in promoting PDAC progression during PDAC immune evasion by coordinating the function and number of immune cells, especially galectin-1. In this paper. we review the involvement of galectins in the construction of PDAC privileged zones by regulating relevant immune cells, establishing fibrotic barriers, and promoting cellular metabolism.

**Abstract:**

Pancreatic ductal adenocarcinoma (PDAC) is one of the most lethal cancers and is highly immune tolerant. Although there is immune cell infiltration in PDAC tissues, most of the immune cells do not function properly and, therefore, the prognosis of PDAC is very poor. Galectins are carbohydrate-binding proteins that are intimately involved in the proliferation and metastasis of tumor cells and, in particular, play a crucial role in the immune evasion of tumor cells. Galectins induce abnormal functions and reduce numbers of tumor-associated macrophages (TAM), natural killer cells (NK), T cells and B cells. It further promotes fibrosis of tissues surrounding PDAC, enhances local cellular metabolism, and ultimately constructs tumor immune privileged areas to induce immune evasion behavior of tumor cells. Here, we summarize the respective mechanisms of action played by different Galectins in the process of immune escape from PDAC, focusing on the mechanism of action of Galectin-1. Galectins cause imbalance between tumor immunity and anti-tumor immunity by coordinating the function and number of immune cells, which leads to the development and progression of PDAC.

## 1. Introduction

Pancreatic ductal adenocarcinoma (PDAC) is the most malignant tumor, accounting for 85% of all pancreatic malignancies [1]. Pancreatic cancer is the third deadliest cancer globally, with a five-year survival rate of 9% and a poor prognosis [2]. Risk factors for pancreatic cancer are smoking [3], alcohol abuse [4], obesity, diabetes, and pancreatitis. PDAC possesses a unique immunosuppressive tumor microenvironment characterized by a low mutational load, a high number of functionally abnormal immune cells, and generally poor infiltration of effector T cells [5,6].

Galectins are a family of soluble proteins consisting of carbohydrate-recognition domains (CRDs) with one or two structural domains. They are divided into three groups based on their CRD structures, i.e., prototypical galectins, tandem repeat galectins, and chimeric galectins. The prototypical galectins possess two identical CRD structures, which include galectin -1, -2, -5, -7, -10, -11, -13, -14, and -15. The tandem repeat galectins are comprised of two different CRD structures, including galectin -4, -5, -8, 9, and -12, and the chimeric galectins consist of only one member, i.e., galectin-3, which has amino acids attached at the N terminus to a single CRD structure [7]. Among the galactoprotein family, galectin-16, as a special member, differs from other galactoproteins in that galectin-16 lacks the ability to bind β-galactose. Galectin-16, as a monomeric structure, has different biological roles, and the current study shows that galectin-16 promotes apoptosis of T cells by binding to c-Rel [8]. There are 12 galectins present in humans, while galectin-5, -6, -11, and -15 are not expressed in humans. Galectins perform various biological functions, such as inducing apoptosis, promoting intercellular adhesion or tumor cell-extracellular matrix adhesion, mediating intercellular signaling, promoting angiogenesis, enhancing cellular metabolism, and tumor immune escape. In tumor cells, galectins promote tumor cell proliferation and metastasis by enhancing the expression of oncogenic signals and helping tumor cells complete immune escape [9,10]. Galectin-1 and galectin-3 are highly expressed in several tumor tissues, with the former being emerged as a new target for clinical trials. Galectin-1 is a highly conserved β-galactoside-binding protein [11], which is encoded by the LGALS1 gene [12]. In PDAC, galectin-1 is chiefly secreted by pancreatic stellate cells (PSC) and PDAC cells. Under normal conditions, PSC cells are in a static state, but once stimulated by pancreatitis, infection, and other stimuli, they get activated and oversecrete galectin-1, hence, contributing to high fibrosis of PDAC [13]. Galectin-1 has a wide range of biological functions that are involved in tumor cell transformation, angiogenesis, immune evasion, sensitivity to radiotherapy, and regulation of the cell cycle, apoptosis, and inflammation [14]. Galectin-1 is considered an immunomodulatory protein involved in the body’s normal immune response and the development of cancer by regulating the function of immune cells [15,16,17,18]. The immunosuppressive effect of galectin-1 in PDAC has been linked to T cell apoptosis. Meanwhile, galectin-1 interacts with various other malignant immune cells, which play a pro-cancer role in PDAC [19]. In this paper, we review the involvement of galectins in the construction of PDAC privileged zones by regulating related immune cells, establishing fibrotic barriers, and promoting cellular metabolism.

## 2. Galectin-1 Is Involved in the Immune Evasion of PDAC

Galectin-1 is important in mediating the immune evasion of tumor cells. In PDAC, it forms an immunosuppressive microenvironment by expressing immunosuppressive cells and suppressing the activity of T cells. Meanwhile, galectin-1 also interacts with cancer-associated fibroblasts (CAFs) and PSCs, creating a fibrotic barrier that blocks the entry of immune drugs and immune-related cells. Finally, galectin-1 enhances immune evasion of tumor cells by promoting angiogenesis and enhancing cellular metabolism around tumor tissues.

### 2.1. Galectin-1 Participates in the Immune Escape of PDAC by Forming an Immunosuppressive Microenvironment

There are five primary mechanisms of immune escape in cancers, which are tumor-induced immunosuppression, tumor induction immunity area, low immunogenicity, recognition as an autoantigen, and antigenic modulation [20]. Tumor-induced immunosuppression and tumor induction immunity act as a lead in PDAC. Galectin-1 is involved in immune regulation, which can induce the apoptosis of T cells [21], NK cells, and other immune cells [10,22]. High expression of galectin-1 often leads to abnormal function or inactivation of immune cells, promoting immune escape of tumors. Normal immune cells such as B cells, T cells, macrophages, NK cells, and dendritic cells (DC) are suppressed in PDAC patients, and TAM and myeloid-derived suppressor cells (MDSC) are heavily recruited into the tumor microenvironment, all of which interact with galectin-1, and help the tumor escape immune attack.

#### 2.1.1. Galectin-1 Induces the Production of TAM Cells

In cancer, the host will recruit and activate all types of cells to resist, including the infiltration of T cells, the drive of macrophages, and the activation of NK and DC cells. However, as cancer progresses, activated immune cells replace anti-tumor as “agents” that promote tumor development, particularly TAM cells, which induce apoptosis or dysregulation of T cells while inhibiting the function of normal immune cells and promoting the development and metastasis of tumor cells [23].

TAM, an M2-type macrophage that participates in tumor immune escape, has been implicated in many cancers, including PDAC [24,25]. TAM and MDSC inhibit the activity of T cells and induce T cell apoptosis [26,27]. High expression of TAM is associated with hypoxia-inducible factor-1 (HIF-1) activation and lactate production. One of the characteristics of PDAC is hypoxia, which plays an important role in the generation and displacement of TAM [28]. Under hypoxic conditions, HIF-1 signaling enhances galectin-1 expression, which promotes the transcriptional activity of HIF-1 via H-ras [29]. In this hypoxic environment, MDSC converts to TAM, permitting the formation of a tumor immunosuppressive microenvironment. On the other hand, lactate induces the expression of HIF-1, which converts MDSC to TAM. Lipopolysaccharide (LPS) induces galectin-1 to produce ADAM10/17, promoting lactic acid secretion [30,31,32,33]. Large amounts of secreted lactate can activate nuclear factor erythroid 2-related factor 2 (Nrf2), which regulates the production of reactive oxygen species (ROS), leading to the differentiation of TAM and the expression of vascular endothelial growth factor (VEGF) [34]. VEGF is vital in encouraging PDAC tumor immune evasion for MDSC to enter tumor cells. Galectin-1 stimulates the expression of VEGF by promoting IL-6 production. Galectin-1 secreted by PSC results in over-secretion of Th2 cytokines by T cells, including interleukin (IL-6). Highly secreted IL-6 causes high MDSC expression through the IL-6/Janus kinase (JAK)/STAT3 pathway. Neuropilin1 (NRP-1) is a galectin-1 binding site on the surface of CAFs [35]. NRP-1 promotes vascular endothelial growth factor receptor2 (VEGFR2) signal transduction induced by galectin-1 [36]. Meanwhile, VEGFR2/NRP-1 trans-complex reduces angiogenesis and aggravates the hypoxic environment of PDAC [37,38], which promotes the activation of TAM and MDSC. Interestingly, IL-8 secreted by CAFs can promote high expression of NRP-1, while in PDAC, galectin-1 can induce more IL-8 secretion by CAFs [39]. Galectin-1 may promote NRP-1 expression by activating IL-8 secretion, and then galectin-1 binds to NRP-1 that is highly expressed on the surface of CAFs or PSC, ultimately promoting PDAC tumor immune evasion. Galectin-1 promotes the role of TAM in PDAC immunosuppression by participating in the induction and activation of TAM cells (Figure 1).

#### 2.1.2. Galectin-1 Inducing Dysfunction and Death of T Cells in PDAC

The T cells are regarded as an important defense entity with anti-tumor immunity; after receiving APC signals, they are activated and differentiated into cytotoxic T lymphocyte (CTL) cells, killing and regulating tumor cells. Tumor cells in cancer “fight back” against T-cell killing by inducing abnormal T-cell function or reducing the number of T-cells. T cell apoptosis and the infiltration of inhibitory Treg cells are essential for immune escape in PDAC. Galectin-1, a known negative regulator of T cells, is highly expressed in PDAC, where it promotes the progression of cancer by inducing abnormal T cell function and apoptosis [40,41].

The common T cell apoptosis pathways are caspase-mediated and caspase-independent, where the former is common in PDAC. The caspase-3, caspase-8, and caspase-9 are highly expressed in PDAC and are involved in the apoptosis of T cells induced by galectin-1 [42]. Caspase-3 acts as an “executor” in cell apoptosis. Fodrin is a cytoskeletal molecule that binds to CD45 [43] and is the primary target of lytic caspases, particularly caspase-3, whose cleavage results in membrane dysfunction and cell abnormality. CD45 is an indispensable receptor for galectin-1 and a critical component of galectin-1-mediated T-cell apoptosis [44]. The binding of galectin-1 and CD45 degrade fodrin, which further enhances the phagocytosis of galectin-1-treated T cells by macrophages and promotes T-cell apoptosis [45]. Galectin-1 increases the sensitivity of dormant T cells to fas/caspase-8-mediated cell death [46]. Galectin-1 induces T cell apoptosis in PDAC via the caspase-8/Fas pathway, high expression of caspase-8, galectin-1, and poor prognosis of patients support this hypothesis [46,47]. Galectin-1 initiates sphingomyelinase-mediated ceramide release [48]. The activation of caspase-9 and caspase-3 depends on the surge of ceramide levels. In PDAC, ceramide and its metabolites are highly expressed [49,50,51], and galectin-1 promotes T cell apoptosis by releasing ceramide and activating caspase-9 and caspase-3. In low concentrations of galectin-1, the caspase-dependent pathway induces apoptosis, characterized by caspase-3 and caspase-9 activation. The p56lck and Zeta-chain-associated protein kinase-70 (ZAP-70) are the most critical in this process. The P56lck is expressed in all T cells, while ZAP-70 is positively correlated with T cell infiltration. Notably, galectin-1 induces T cell apoptosis in conjunction with ZAP-70 and p56lck in PDAC. The caspase non-dependent pathway can mediate apoptosis under high concentration of galectin-1 [52]. In PDAC, the low expression of galectin-1 first activates caspase-3, caspase-9, and then caspase-8, resulting in the promotion of T cell apoptosis through different mechanisms.

CD7 is another receptor of galectin-1 [53], which catalyzes apoptosis of T cells induced by galectin-1 [54,55,56]. Two pathways influence CD7 expression, the activation of nuclear factor kappa-B (NF-κB), which regulates CD7 expression in T cells, and high expression of SP1, which promotes CD7 expression [57]. In PDAC, galectin-1 induces high expression of CD7 via high expression of SP1 and activation of NF-κB [44,58]. High expression of galectin-1/CD7 promotes further apoptosis of T cells. Macrophage galactose-type lectin (MGL) is usually expressed on the surface of immature DC cells and is linked to immune escape. In PDAC, galectin-1 and MGL synergistically induce apoptosis of T cells. In addition, high expression of c-Jun N-terminal kinase (JNK)/CJun/activator protein 1 (AP-1) in PDAC induces apoptosis of T cells, and galectin-1 promotes T cell apoptosis through activation of the JNK/C-Jun/AP-1 pathway, promoting PDAC tumor immune evasion [59].

Tumor-associated complements, such as complement C3 and C5 and their metabolites, are highly expressed in PDAC [60]. IL-4 is highly expressed in PDAC [61], and under the induction of IL-4, galectin-1 is released in large quantities by macrophages [62]. Complement receptor 3 (CR3) binds to specific ligands, such as IL-4, to induce the “outside-in” signaling pathway of macrophages, leading to the production of IL-1 and IL-6 and phagocytosis of macrophages. Galectin-1 improves CR3 function and induces T cell apoptosis through two modes [63]. The first model is that galectin-1 promotes the binding force between CR3 and the ligand; the second model builds a bridge between CR3 and CR3-associated receptors and activates CR3 through the above two modes. Meanwhile, galectin-1 enhanced the activity of CR3, resulting in macrophage activation, and accelerated T cell apoptosis. The role of galectin-1 in the induction of T cell apoptosis is summarized in Figure 2.

#### 2.1.3. Galectin-1 Destroys the Normal Function of NK Cells in PDAC

NK cells participate in autoimmunity by secreting IL-1, IL-5, IL-8, IL-10, tumor necrosis factor-α (TNF-α), and Interferon-γ (IFN-γ) to kill aging, virus-infected, and tumor cells. Increased NK cell infiltration in cancer suggests a poor prognosis, and the conceivable mechanism is that NK cells become functionally impaired to exert their normal anti-tumor effects. Abnormal NK cells are present in large numbers in PDAC and are often associated with hyperglycemia. The galectin-1 promotes immune evasion of PDAC tumor cells by interacting with NK cells and inhibiting normal NK cell function.

NK cells are activated by IL-2, and the release of IFN-γ and TNF-α play a role, but all three have lower expression in PDAC [64], prompting an anomaly in pancreatic cancer. In PDAC, galectin-1 can directly inhibit the production of IL-2 and disturb the Th1/Th2 balance, following a subsequent decrease in Th1 cells and the secretion of cytokines, such as IL-2 [65], thus, reducing the activation of NK cells. In addition to IL-2, IL-6 is also involved in the negative regulation of NK cells, and galectin-1 promotes the production of IL-6, which is highly and negatively correlated with NK cell activity. Remarkably, high expression of galectin-1 was positively correlated with adipocyte infiltration. While obesity is an important risk factor for PDAC, it has been observed that the obese mice were having a higher incidence of pancreatic cancer than normal mice, with peripancreatic infiltration with adipocytes, high IL-6 expression, reduced IFN-γ secretion, and inhibition of NK cell activity [66]. In pancreatic cancer, galectin-1 promotes the secretion of matrix metalloproteinase (MMP9) and IDO [67], and high expression of MMP9 and IDO can result in NK cell dysfunction [65]. NK Group 2, Member D (NKG2D) is expressed on the surface of NK cells and combines with MHC class I-related molecule A (MICA) and MICB to activate NK cells and participate in the biological functions of NK cells. NK cells from healthy people can kill PDAC cells under the mediation of NKG2D, but in PDAC, NKG2D is low in expression, making NK cells unable to clear tumor cells [68]. Galectin-1 forms the hypoxic microenvironment and contributes to the immune evasion of PDAC tumor cells by enhancing HIF activity, promoting the formation of ADAM10, inducing the low expression of NKG2D, and reducing the activity of NK cells in PDAC, which results in poor proliferation of NK cells [69].

Furthermore, hypoxia can also induce the upregulation of HIF-1 and metalloproteinase domain 10 (ADAM10) [32], leading to decreased NKG2D of NK cells and enabling tumor cells to escape immune monitoring and NK cell-mediated lysis. Most pancreatic cancer patients have hyperglycemia and type-2 diabetes. Diabetes mellitus is a risk factor for PDAC, and vice versa. As a marker for type-2 diabetes, galectin-1 production is associated with diabetes. Type 2 diabetes is characterized by hyperglycemia, insulin resistance, and hyperinsulinemia. Insulin resistance is common in PDAC patients and is further aggravated by galectin-1 [70]. Moreover, the damage caused by pancreatic cancer to the islets of Langerhans also aggravates insulin resistance. Compensatory insulin reversely promotes PSC activation [71], leading to further increased galectin-1 secretion. The AMPK-BMI1-GATA2-MICA/B pathway is activated due to hyperglycemia, which triggers the inactivation of MICA [72]. As a member of NKG2DL, the inactivation of MICA leads to the dysfunction of NK cells. Galectin-1 inhibits the normal function of NK cells via multiple mechanisms, ultimately promoting PDAC tumor immune evasion (Figure 3).

### 2.2. Galectin-1 Enables Tumor Cells to Gain Immune Privileges by Remodeling the Extracellular Matrix

The deposition and cross-linking of the extracellular matrix will induce the development of fibrosis, and the sclerotic matrix will inhibit the entry of normal immune cells and promote the growth of malignant tumors. The immune privilege of tumor cells is apoptosis, inactivation of immune cells, and a highly fibrotic barrier of the stroma surrounding the tumor. According to research, immune cells cannot penetrate tumor tissue to exert anti-tumor effects due to the fibrotic barrier. The tumor cell stroma in PDAC is highly fibrotic, making it difficult for immune cells and chemotherapeutic agents to penetrate and exert their effects [73]. Moreover, the fibrotic barrier prevents immunocompetent cells from entering the tumor and prevents renegade cells from leaving the tumor, ultimately promoting an immunosuppressive microenvironment. Galectin-1 promotes the deposition of associated proteins by recruiting PSC and CAF into the extracellular matrix, thereby remodeling the extracellular matrix around PDAC, ultimately promoting the formation of a fibrotic barrier [74,75].

#### 2.2.1. Galectin-1 Activates CAF and Promotes PDAC Fibrosis

CAF has been identified as a cell that promotes tumor fibrosis and accounts for a significant proportion of extracellular matrix (ECM) in pancreatic cancer. CAF is activated by transforming growth factor-β (TGF-β), IL-6, and IL-10 [76,77,78,79]. MMPs are secreted and activated after CAF activation to promote basement membrane (BM) degradation, thereby remodeling ECM. Additionally, CAF is involved in fibrosis through hyaluronic acid and collagen secretion. In PDAC, galectin-1 is an important pro-fibrotic substance that acts by regulating the activation of CAF. Galectin-1 stimulates IL-6, and IL-10 secretion in PDAC activates CAF expression and promotes stromal fibrosis of the tumor [75]. In mice experiments, researchers found that galectin-1 promotes CAF activation through the activation of hedgehog (Hh). Another study found that PSC high expression of galectin-1 indirectly promoted the CAF activation through upregulation of TGF-β expression. These activated CAFs promote MMP secretion and are involved in extracellular matrix remodeling.

#### 2.2.2. Galectin-1 Is Involved in PSC-Mediated PDAC Fibrosis

PSC is the main driver of stromal fibrosis in PDAC and plays a key role in remodeling the extracellular matrix. A large number of studies have now shown that PSC secretes IL-1, IL-6, IL-8, IL-10, VEGF, platelet-derived growth factor (PDGF), FAP, Hh, MMP, and TGF-β [80], which are directly involved in collagen synthesis or differentiation into CAF cells, and are part of the “fibrotic network” by cleaving the fibrous matrix to promote ECM deposition core. Galectin-1, a product of PSC secretion, has gained attention for its regulation and synergistic effect. In in vitro experiments, researchers found that adding galectin-1 to PSC culture could induce the proliferation of stellate cells, accompanied by the synthesis of collagen [81]. In addition, serum factors secreted by PSC could increase galectin-1 secretion, suggesting that galectin-1 and PSC promote and synergize each other. Galectin-1 induces the secretion of IL-6 and IL-10 by PSC and enhances the secretion of IL-8 by CAF, promoting extracellular matrix deposition [82]. In a mouse model, galectin-1 was reported to promote IL-10 secretion by PSC and induce fibrotic tumor stroma formation [65]. Under hypoxic conditions, VEGF recruits CAF and PSC into tumor tissues and promotes tumor tissue fibrosis [83]. Galectin-1 induces the formation of a hypoxic microenvironment in PDAC by promoting the expression of HIF, which causes fibrotic effects in PSC [84]. Furthermore, pancreatic cancer tumors in mice with galectin-1 deficiency exhibited abnormalities in the stroma related to the Hh signaling pathway [75]. Galectin-1 could directly activate the Hh pathway in the tumor stroma and promote the fibrosis of PDAC. The Hh signal is a chemotactic signal for PSC, which can recruit PSC into the tumor and promote collagen and fibronectin expression [85]. MMP is an enzyme responsible for fibrinolysis, and tissue inhibitor of matrix metalloproteinases (TIMP) is an inhibitor of MMP. Overexpressed galectin-1 in PSC promotes the increase in MMP and TIMP through the TGF-β/SMAD pathway, but the changes in MMP are less than that of TIMP. Resultantly, TIMP inhibits the effect of MMP dissolving ECM and promotes fibrosis [80].

One of the causes of fibrosis in the tumor is inflammation. In PDAC, inflammation can induce PSC activation and promote fibrosis [86]. In the presence of the proinflammatory factor NF-κB, the apoptosis of PSC is inhibited, and the secretion of TIMP is increased. Galectin-1 is a vital NF-κB activator, and galectin-1 expressed by PSC is involved in the activation and secretion of NF-κB [87], which promote each other and ultimately results in fibrosis of the extracellular matrix. Moreover, PSCs interact with macrophages and B lymphocytes in the tumor and promote the formation of PDAC stromal fibrosis [88,89]. Galectin-1 increased the binding capacity of CR3, ensuring an increase in the action strength of IL-4. The IL-5 is an inflammatory cytokine and novel pro-fibrotic factor that recruits B cells and eosinophils into tumors. It is a Th2 type of immune response that ultimately promotes the formation of stromal fibrosis in tumors [90]. Galectin-1 breaks the Th1/2 balance in pancreatic cancer, and the over-expression of galectin-1 in PSC may enhance the release of IL-5, thus, enhancing the tumor fibrosis [65]. The release of IL-5 induces the infiltration of B cells in PDAC, which promotes the activation of PSC and collagen production through the secretion of PDGF-B [91]. Galectin-1 stimulates the release of the inducer IL-5 in PDAC while recruiting B cells via the release of IL-6, BTK, and other substances. As a result, galectin-1 creates favorable conditions for B cells to cause fibrosis in PDAC. Galectin-1 enhances fibrosis of PDAC tumor stroma through interaction with tumor-associated cells (Figure 4).

### 2.3. Galectin-1 Promotes Immune Evasion of Tumor Cells through Other Adjuvant Modalities

Abundant vascular tissue provides nutrients and oxygen to tumor cells while carrying away unwanted metabolic wastes, promoting proliferation and metastasis of tumor cells to a great extent. In PDAC, the vascular distribution shows high density, poor perfusion, and impaired integrity [92]. The prominent feature of PDAC is the presence of large deposits of fibrous interstitial fluid, which generates high interstitial fluid pressures, compressing the vasculature, and inevitably leading to reduced immune drug penetration and uptake [93]. Angiogenesis is dependent on the interaction of the cellular and extracellular microenvironment, and galectin-1 and its other family members play a critical role in it [94]. In PDAC, the tumor cells satisfy their own nutritional needs; their metabolites promote the production of immunosuppressive cells while suppressing the production of CD8+ T cells [95]. Galectins contribute to the metabolic process of tumor cells, indirectly promoting immune evasion of tumor cells [96].

Abnormal energy metabolism is the characteristic change in tumor tissue compared to normal tissue [97]. Tumor tissues undergo metabolic reorganization to obtain the necessary energy and are more susceptible to glycolysis even at high oxygen concentrations, called the “Warburg effect” [98]. In the specific metabolic process of tumor cells, galectin-1 participates and facilitates metabolic reactions. The hypoxic microenvironment facilitates the metabolism and growth of tumor cells. HIF-1α regulates the expression of galectin-1. In the hypoxic microenvironment, stable HIF-1α induces the expression of galectin-1 and glucose transporter-1 (GLUT1), which promote angiogenesis, tumor proliferation, and metastasis [99,100]. Meanwhile, other members of the galectin family play similar roles, for example, high expression of galectin-3 promotes the expression of GLUT1 through the PI3K signaling pathway, which in turn enhances glycolysis in tumor tissues [101,102].

Furthermore, lactate, a byproduct of glycolysis, creates an acidic environment that encourages angiogenesis and the recruitment of immunosuppressive factors and cells [103]. A TLR4 ligand lipopolysaccharide induces high expression of galectin-1, which in turn, accelerates the activation of glycolysis-related enzymes, such as hexokinase (HK), phosphofructokinase (PFK), and lactate dehydrogenase A (LDHA), promoting lactate production [96,104]. Galectin-1 and galectin-3 tumors enhance adaptation to the hypoxic microenvironment by promoting angiogenesis, conversion of tumor cell metabolism to glycolysis, and tumor cell adaptation to metabolic stress [105,106]. Galectin-1 and other galectins ultimately stabilize the microenvironment for tumor cell growth by participating in specific metabolic processes of tumor cells: meeting the metabolic demands of tumor cells, inhibiting the function of immune cells, and creating an immunosuppressive microenvironment.

## 3. Galectin-3 and Galectin-9 Promote Immune Evasion of PDAC Tumor Cells

Galectin-3 is a structurally unique beta-galactoside-binding protein that has important roles in tumor proliferation and metastasis. Specifically, galectin-3 is not detected in normal pancreatic organs but is highly expressed in pancreatic cancer patients [107]. In contrast to galectin-1, the galectin-3 helps pancreatic tumor cells to participate in immune evasion mainly by interacting with the immune cells. Galectin-3 is mainly released by PDAC cells. First, galectin-3 aggregates T cell antigen receptors (TCRs) on the cell surface by binding to the TCR receptor and inhibiting its function. It directly acts on related glycoprotein receptors, such as CD71 and CD45, to inhibit T cell activity and increase activity apoptosis of T cells [55]. The interaction of galectin-3 with T cell-expressed α3β1 integrin inhibits T cell proliferation and promotes the formation of a PDAC immunosuppressive tumor microenvironment [108]. Galectin-3 inhibits IFN-γ secretion by lymphocytes. Knockdown of galectin-3 on the surface of CD4+ T cells resulted in a substantial increase in IFN-γ secretion by lymphocytes [109]. Demotte et al. found that in vitro tumor cells of pancreatic cancer cultures with high expression of galectin-3 had higher suppression levels of CD8+ T lymphocytes because IFN-γ secretion was substantially reduced. Kouo et al. also reported that in pancreatic cancer patients, the lower the number of lymphocytes surrounding tumor cells with high galectin-3 expression, the lower the patient survival and quality of life [110,111]. Second, galectin-3 is also involved in macrophage differentiation. Macrophages highly express galectin-3, and IL-4/IL-13 promotes galectin-3 expression by mediating the activation of M2 macrophages [112]. Similarly, galectin-3 activates M2 macrophages by binding to glycoprotein receptors, such as CD98, and triggering the activation of PI3K [113]. Galectin-3 can also promote the proliferation and expression of M2 macrophages by converting M1 macrophages into M2 macrophages. By interacting with galectin-3, M2 macrophages suppress the systemic immunity and promote immune evasion of PDAC cells. Song et al. discovered that galectin-3 is highly expressed on the surface of PDAC cells, which suppressed the systemic immune system function and promoted tumor proliferation and metastasis by activating the RAS signaling pathway [114]. The tumor microenvironment of PDAC is characterized by marked hypoxia and starvation. The PDAC tumor cells promote galectin-3 expression, reduce infiltration of associated lymphocytes, adapt to the tumor microenvironment under conditions of hypoxia and starvation, and promote further tumor cell development [115].

Galectin-1 and galectin-3 are important contributors to the regulation of immune function and have a unique dual role in tumor regulation. Extracellular galectin-1 and galectin-3 play immunosuppressive roles by promoting T cell apoptosis, while intracellularly they inhibit apoptosis and promote T cell proliferation [116,117]. Galectin-1 and galectin-3 assume key roles in stabilizing immune function in addition to playing a critical role in tumor immune evasion [118]. Galectin-1 stabilizes autoimmune function by downregulating pro-inflammatory cytokine expression and promoting IL-10 secretion, inhibiting deleterious Th1 responses and enhancing immune cell resistance [119]. Intracellularly, galectin-3 stabilizes autoimmune function by negatively regulating the onset of inflammatory responses. Galectin-3 negatively regulates the onset of inflammatory responses mediated by LPS by binding to LPS and inhibiting the production and release of inflammatory factors [120]. In addition, intracellularly, galectin-3 inhibits apoptosis and promotes T cell proliferation. Galectin3 binds to Bcl-2 and inhibits T cell apoptosis in a lactose-inhibitory manner [121]. Galectin-3 also protects T cells from apoptosis by stabilizing the structure and function of mitochondria and inhibiting apoptotic toxins and aggression [122]. In addition to promoting the proliferation of T cells, intracellular galectin-3 also attracts and induces the expression of other immune cells. Hsu et al. demonstrated that in a mouse model, galectin-3 deficient macrophages were more apoptosis sensitive and immunocompromised [123]. Furthermore, it was found that galectin-3 is essential for phagocytosis and is involved in phagocytosis through a number of mechanisms within the cell [124]. It mainly includes phagocytosis of microorganisms and apoptotic cells and promotes the expression of immune functions. Galecitin-3 was found by Sano et al. to act as a novel chemoattractant, attracting monocytes and phagocytes through a partial PTX pathway, eliciting a strong immune response [125].

Galectin-9 is similar to other galectins and has multiple biological functions. However, in cancer, galectin-9 both promotes tumor development and inhibits tumorigenesis and transformation, depending mainly on the binding of galectin-9 to T cells and other tumor cell surface receptors [126]. It was summarized that galectin-9 promotes tumor development in pancreatic cancer by inhibiting immune cell activity. In pancreatic cancer, galectin-9 binds to dectin-1 on macrophages and suppresses immune cell activity by reprogramming CD4+ and CD8+ T cells, promoting the formation of an immunosuppressive microenvironment in PDAC [127]. Galectin-9 helps tumor cells accomplish immune evasion by interacting with TIM-3 expressing Th1 cells and promotes their apoptosis [128]. Galectin-9 promotes an immunosuppressive environment by interacting with 4-1BB, CD44, and TIM-3 expressed on the surface of T cells, inducing T cell apoptosis, and inhibiting T cell proliferation [10]. Moreover, it was found that galectin-9 knockdown in tumor cells in PDAC enhanced the activity of T cells in the tumor microenvironment, and inhibited tumor growth [129]. In immunotherapy, a study by Yazdanifar et al. confirmed that blocking galectin-9 expression enhances the activity of relevant T cells in CAR-T therapy and inhibits tumor progression [130].

Galectin-9 also has a dual role in tumors, with TIM-3 being the ligand that can bind highly to galecitin-9. TIM-3 is one of the immune checkpoint proteins, and the binding of galectin-9 and TIM-3 plays a double-edged role in immune function [131]. This dual role depends mainly on the cellular phenotype of TIM-3. TIM-3 activates NK cell expression and promotes IFN-γ secretion by binding galectin-9, thus, enhancing immune function [132]. In contrast, in T cell subsets (Th1, Th17, Tc1), the combination of TIM-3 and galectin-9 can inhibit T cell activity, suppress the production of TNFα, IFNγ, and IL-17, induce T cell apoptosis and, thus, mediate the development of immunosuppression [133]. It often plays a negative regulatory role in anti-tumor immunotherapy. Since TIM-3 is highly expressed on CD8+ T and FoxP3 Treg cells, both of which assume key roles in the formation of tumor immunosuppression, blocking the TIM-3/galectin-9 pathway is of great interest [134]. The use of anti-TIM-9 monoclonal antibodies will be an important direction in immunotherapy. In addition, it was found that PD-1/PD-L1, which has received the most attention, does not work very well in solid tumors using a single anti-PD-1 or antiPD-L1, and elevated expression of TIM-3 was observed after its use. Galectin-9 was also able to bind to PD-1 and enhance the surface TIM-3 and PD-1 expression of T cells [135]. Similarly, Limagne et al. found that the immune checkpoint pathway, such as TIM-3/galectin-9, was upregulated after the anti-PD-1 drug Nivolumab, and most patients showed a high degree of drug resistance [136]. Therefore, combined blockade of TIM-3/galectin-9 and PD-1/PD-L1 pathways is of great importance in anti-tumor immunotherapy. Several clinical trials are also working on combined blockade of TIM3/galectin-9 and PD-1/PD-L1 pathways (NCT02817633, NCT03099109, NCT02608268), and other clinical trials focusing on the combination of anti-TIM3/galectin-9 and anti-PD-1/PD-L1 in tumor immunotherapy will be the new strategy. In PDAC, Dectin-1 is a C-type lectin receptor in macrophages, and galectin-9 is a ligand for Dectin-1. The combination of the two can induce T cell apoptosis and facilitate the development of immunosuppression. The combined blockade inhibitor of galectin-9/Dectin-1 and PD-1 can promote T cell activity and reduce T cell apoptosis, thus, enhancing antitumor immune function [137].

## 4. The Role of Other Galectins in Immune Evasion of PDAC

The other galectins also have a role in helping tumor cells to participate in immune evasion. In the tumor cell model with high galectin-7 expression, the number of T cells was found to significantly decrease, which cemented the hypothesis that galectin-7 could directly induce T cell apoptosis [138]. Galectin-8 also has a dual role in tumor immunomodulation. Galectin-8 is mainly secreted by vascular endothelial cells and lymphocytes. Galectin-8 is currently limited in pancreatic cancer, but in other solid cancers, it exhibits potent pro-inflammatory effects, suppresses immune cell function in tumor cells, and promotes tumorigenesis [139]. Galectin-8 enhances the immune response by inducing the proliferation of pure CD4+ T cells and promoting the secretion of IL-2, IL-4, and IFN-γ by T cells as well as activating the proliferation of B cells that release IL-6 and IL-10, which together mediate the immune response. Galectin-4 is mainly secreted by epithelial tumor cells and less frequently expressed by stromal cells [140,141]. Galectin-4 is highly expressed in pancreatic cancer while almost not in normal tissues. This suggested a special significance of galectin-4 in pancreatic cancer. Galectin-4 acts as an inhibitor of immune evasion of tumor cells in pancreatic cancer [110]. A study discovered that high expression of galectin-4 was negatively correlated with lymphatic metastasis of pancreatic cancer and positively linked with T cells in vivo [142]. Galectin-4 enhances immune function by binding to CD14 and promoting MAPK through expression to promote macrophage formation [143]. Here, the role of different galectins in pancreatic cancer immune evasion was reviewed (Table 1).

## 5. The Value of Galectins in the Diagnosis and Treatment of PDAC

The clear differences in the expression of various galectins in patients with pancreatic cancer and normal subjects also suggest a clear diagnostic significance of galectins. A meta-analysis showed that high expression of galectin-1 in PDAC tumor tissue was associated with poor prognosis in pancreatic cancer patients, while high expression of galectin-9 and galectin-4 had a good prognosis [144]. In the plasma of PDAC patients, galectin-3 was highly expressed in pancreatic cancer patients, with predictive sensitivity and a specificity of 74.8% and 90.2%, respectively, which also translates into shorter survival time and a poor quality of life for patients [145]. The prognosis of other galectin family members and pancreatic cancer could not be determined accurately and had opposite results in different experiments. Furthermore, the combination of galectin-1 and the conventional tumor marker CA19-9 in diagnosing pancreatic cancer improves the accuracy and specificity of the diagnosis. A study by Xie et al. stated that the combination of galectin-3 and other biomarkers, CA19-9 and CEA, greatly improved the accuracy of the diagnosis of pancreatic cancer [107]. A study reported a covalently attached glycosylated peptide derived from tissue plasminogen activator to the surface of nanoparticles for actively targeting the galectin-1 as the target receptor. During their evaluation in a mouse model using the nuclear magnetic resonance technique, the results showed significant uptake of those nanoparticles, which offers a novel approach to the diagnosis of PDAC [146]. Multiple bioinformatics analysis has shown that the diagnostic and prognostic efficacy of galectin-1 is significantly higher than that of biomarkers in the proteomic analysis [147,148,149].

Galectin-related drugs have a high value in many solid cancers; unfortunately, no clinical studies are related to PDAC. Based on the efficacy of galectin inhibitors in other solid cancers, galectins have great potential in treating PDAC. Drugs targeting galectins can act in numerous ways, such as enhancing sensitivity to radiotherapy and chemotherapy, anti-angiogenesis, immunotherapy, and against tumor growth and metastasis promoted by the hypoxic microenvironment. Galectin-1 and galectin-3 are the most studied galectins, and their related inhibitors are under development. Here, we focus on the role of different types of galectin-1 and galectin-3 inhibitors in anti-tumor.

The current galectin-1 inhibitors are mainly (1) Thiodigalactoside (TDG), (2) Anginex, (3) OTX008, (4) F8.G7, (5) GM-CT-01 (DAVANAT), or GR-MD-02 [150]. As a synthetic disaccharide, TDG is a target of galectin-1 and is able to inhibit tumor angiogenesis and immunosuppression by binding to galectin-1. TDG mainly prevents galectin-1 from binding to CD44 and CD326 on the surface of tumor cells, inhibiting vascular endothelial cells and neovascularization, while inducing CD8+ T cell production and promoting immune cells to block infiltration into the tumor [151]. Anginex is able to bind specifically to galectin-1 and inhibit tumor cell proliferation and metastasis. Anginex is also able to block galectin-1 from entering endothelial cells and reduce endothelial cell phosphorylation [152]. Anginex, when used with radiotherapy and chemotherapy, also promotes the sensitivity of tumor cells to radiotherapy and chemotherapy. Upreti et al. found that Anginex, when combined with chemotherapeutic agents (arsenic trioxide and cisplatin), had fewer side effects and tumor growth was inhibited with an 80% decrease in growth rate [153]. OTX008, in combination with galectin-1, inhibited tumor cell growth signals by mainly inhibiting the p-ERK 1/2 and p-AKT pathways in tumor cells [154]. Michael et al. found that the combination of OTX008 and rapamycin inhibited tumor growth much more effectively than rapamycin alone [155]. F8.G7 and galectin-1 binding inhibited tumor angiogenesis and tumor growth mainly by blocking the action of galectin-1 and VEGF [156]. DAVANAT, a galactomannan, enhances the anti-tumor response by binding to galectin-1 and is able to promote T cell activity and induce IFN-γ secretion by T cells [157].

The galectin-3 inhibitor mainly consists of G3-C12 and Modified Citrus Pectin (MCP). G3-C12 can induce galectin-3 into immune cells, stabilize mitochondria, and exert anti-apoptotic effects. G3-C12, in combination with chemotherapeutic agents (doxorubicin, 5-fluorouracil), can effectively improve the efficacy of chemotherapy and inhibit tumor cell proliferation. In a mouse model, the tumor growth rate decreased by 81.6% [158]. MCP induces cell cycle arrest and promotes apoptosis of tumor cells mainly by binding to galecitin-3. MCP family members PectaSol-C and GCS-100 are both able to bind to galectin-3, inhibit angiogenesis and immune evasion, and promote tumor cell apoptosis [159]. When combined with chemotherapeutic agents, HUVEC cell activity was inhibited, angiogenesis was reduced, CD8+ and CD4+ T cell function was enhanced, and tumor cell proliferation and metastasis were reduced [160]. Furthermore, HH1-1 is a galectin-3 inhibitor that enhances the body’s anti-pancreatic cancer tumor cell activity by blocking the EGFR/AKT/FOXO3 signaling pathway [161]. Another galectin-3 inhibitor, RN1, inhibits the EGFR/ERK/Runx1 and integrin/FAK/JNK-related signaling pathways by blocking the binding of galectin-3 to the EGFR class and inhibits the proliferation of PDAC cells in vitro as well as in vivo [162]. The current clinical trials of galectin application in solid tumors, which provide a good basis for future clinical application of galectins for the treatment of PDAC, are summarized in Table 2.

## 6. Conclusions

This review stated the possible mechanisms of galectins in helping PDAC cells undergo immune evasion. It focused on the mechanisms related to galectin-1 in helping PDAC cells accomplish immune evasion by forming an immunosuppressive microenvironment, remodeling the extracellular matrix to form a fibrous barrier, and ultimately participating in promoting angiogenesis and tumor cell metabolism. Furthermore, we conclude that galectin-3 and other galectins are involved in immune evasion of PDAC by regulating immune cells. Finally, this paper reviewed the initial progress of galectins in the diagnosis and treatment of PDAC in recent years, focusing on the use in combination with other diagnostic and therapeutic methods. Galectins have good application prospects in diagnosing and treating PDAC and deserve further research.

## Figures and Tables

**Figure 1 cancers-14-05475-f001:**
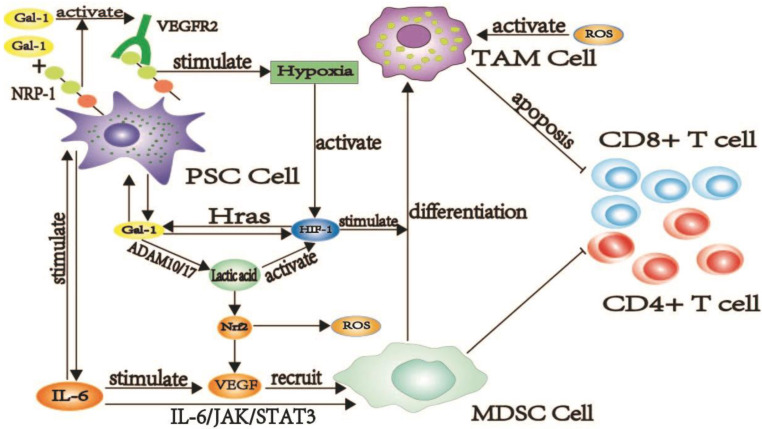
Pathway diagram of galectin-1 involved immunosuppressive signals of PDAC induced by TAM. TAM and MDSC cells prevented PDAC from being killed by T cells by promoting T cell apoptosis and inducing T cell dysfunction. MDSC was a precursor of TAM, and galectin-1 was involved in immunosuppression by promoting infiltration of MDSC and TAM and differentiation of MDSC into TAM. PSC with high expression of galectin-1 secreted more IL-6. IL-6 recruits and activates MDSC cells by promoting VEGF production or through the IL-6/JAK/STAT3 pathway. In addition, IL-6 promotes NRP-1 expression on the surface of PSC, while galectin-1, as the NRP1 receptor, is attracted to the surface of PSC. Under the action of NRP-1, the production of galectin-1-activated VEGFR2 is promoted. Subsequently, VEGFR2 and NRP-1 form trans-complexes, which promote tumor hypoxia. HIF-1 was activated after hypoxia aggravation of PDAC, thus, promoting differentiation of MDSC into TAM. As one of the markers of hypoxia, HIF-1 promotes the transformation from MDSC to TAM. With the participation of H-ras, galectin-1 and HIF-1 promote each other, and both are highly expressed together. Moreover, LPS will induce galectin-1 to secret ADAM10/17, thereby promoting the secretion of lactic acid. On the one hand, lactic acid improves the activity of HIF-1; on the other hand, lactic acid induces the production of ROS and VEGF through Nrf2. VEGF is the recruitment signal of MDSC, while ROS can enhance the immunosuppressive ability of TAM. (This figure was created by ourselves).

**Figure 2 cancers-14-05475-f002:**
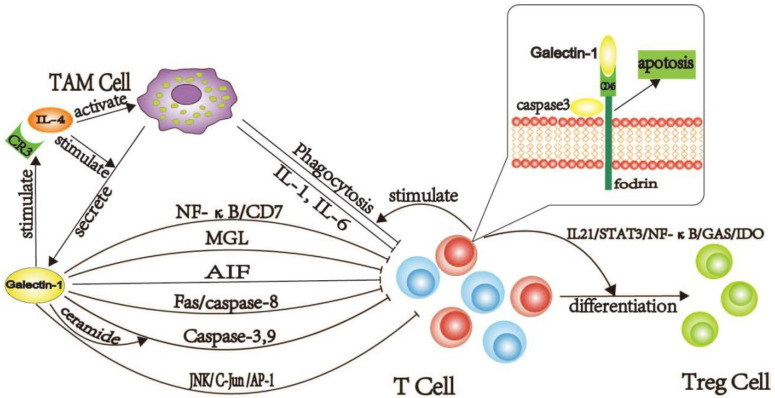
A signal path diagram of the galectin-1 induced T cell apoptosis and dysfunction. By binding to CD7, galectin-1 promoted apoptosis of T cells under the action of NF-κB. In caspase-independent apoptosis, galectin-1 participates in T cell apoptosis by activating AIF. Meanwhile, the stimulation of galectin-1 by Fas/caspase-8, caspase-3, and caspase-9 can contribute to the apoptosis of caspase-dependent cells, and the secretion of galectin-1 can also contribute to the activation of caspase-3 and 9. As a target of caspase-3, fodrin connects CD45 to the cytoskeleton. Under the action of caspase-3, fodrin is lysed rapidly. With the lysis of fodrin, macrophages are more capable of phagocytosis of T cells. Galectin-1 also cooperates with MGL or the JNK/C-Jun/AP-1 pathway to induce apoptosis of T cells. Galectin-1 enhanced the binding force between CR3 and IL-4. IL-4 could promote the secretion of galectin-1 by TAM, activate TAM, and promote the secretion of IL1,6, the T cell-inhibiting factors by TAM. The above effects were also significantly enhanced after the combination of CR3, and IL-4 was enhanced. (This figure was created by ourselves).

**Figure 3 cancers-14-05475-f003:**
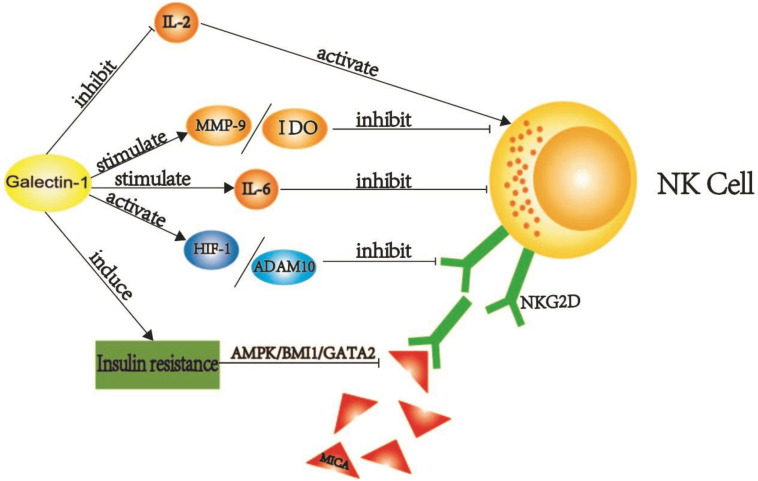
A schematic of galectin-1 induced NK cell dysfunction. Galectin-1 induces the expression of MMP-9, IDO, IL-6, HIF-1, and ADAM10 to inhibit the function of NK cells. Galectin-1 also participates in insulin resistance in PDAC patients and leads to the inhibition of NKG2D ligand MICA through the AMPK/BMI1/GATA2 pathway, which leads to the low function of NKG2D, and then leads to the abnormal function of NK cells. IL-2 is an activator of NK, and galectin-1 is an inhibitor of IL-2, which is involved in the inhibition of IL-2 and leads to low NK function. (This figure was created by ourselves).

**Figure 4 cancers-14-05475-f004:**
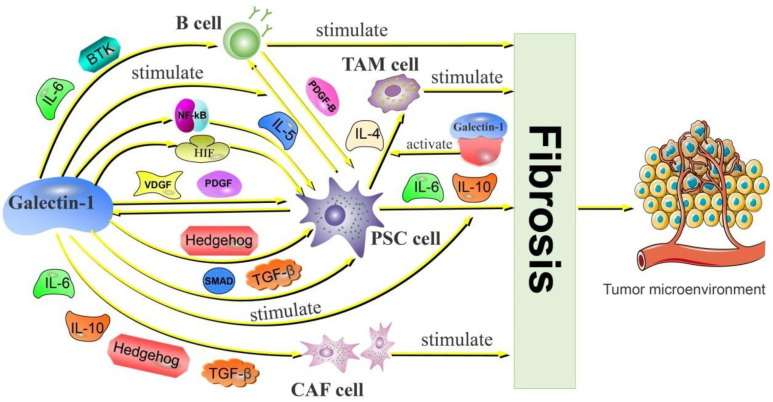
Galectin-1, on the one hand, is involved in the fibrosis induced by CAF cells activated by IL-6,10, HH factors, and TGF-β. On the other hand, galectin-1 activates and recruits PSC cells through HIF, NF-κB, VEGF, PDGF, HH factor and TGF-β/Smad pathways, and participates in the secretion of pro-fibrotic factor IL-6,10 by PSC, and participates in the fibrosis of PDAC through the above pathways. At the same time, galectin-1 promotes the secretion of IL-5 in the PSC, thereby facilitating B cell recruitment. In addition, galectin-1 directly recruits B cells through the release of IL-6 and BTK, and then participates in B-cell-mediated fibrosis. TAM cells are activated by IL-4 secreted by PSC and play a role in promoting fibrosis, while galectin-1 enhances the effect of IL-4 by binding to CR3, leading to further enhancement of TAM cell function. (This figure was created by ourselves).

**Table 1 cancers-14-05475-t001:** Role of different galectins in immune evasion of PDAC cells.

Galectins	Target Cell	Mechanism of Action	Functions	References
Galectin-1	MDSC T cell	Secretion of IL-6 and activation of VEGFR2 Binds to ligands CD2, CD3, CD7, CD45	Induction of TAM cell generation Induction of T cell apoptosis	[31,33] [46,56]
	T cell	Binds to ligands AIF, MGL	Induction of T cell apoptosis	[62]
	NK cell CAF PSCs	Promotes the expression of MMP-9, IDO, IL-6, HIF-1 Secretion of IL-6, IL-10 Secretion of IL-1, IL-6, IL-8, IL-10	Inhibition of NK cell function Promotes fibrotic barrier formation Promotes fibrotic barrier formation	[67,69] [76,77,78,79] [81,82,83]
Galectin-3	T cell	Binds to ligands CD45, CD71	Induction of T cell apoptosis	[54]
	T cell	Binds to ligands TCR	Inhibition of T cell activity	[54]
	M2 type macrophages	Secretion of IL-4/IL-13	Promotes the activation of M2 macrophages	[111]
	M2 type macrophages	Binds to ligands CD98	Promotes the activation of M2 macrophages	[112]
Galectin-9	Macrophages	Binds to ligands dectin-1	Inhibition of T cell function	[124]
	T cell	Binds to ligands 4-1BB, CD44, TIM-3	Induction of T cell apoptosis	[125]
Galectin-7 Galectin-4	T cell T cell	Unknown Unknown	Induction of T cell apoptosis Promotes T cell proliferation and infiltration	[135] [109]

**Table 2 cancers-14-05475-t002:** Galectins are used in various clinical trials for solid cancer treatment.

NCT Number	Title	Status	Conditions	Interventions	Characteristics
NCT03488134	Predicting Prognosis and Recurrence of Thyroid Cancer Via New Biomarkers, Urinary Exosomal Thyroglobulin and Galectin-3	Active, not recruiting	Thyroid Cancer		
NCT04948437	Urinary Exosomal Biomarkers of Thyroglobulin and Galectin-3 for Prognosis and Follow-up in Patients of Thyroid Cancer	Recruiting	Thyroid CancerPapillary Thyroid CancerFollicular Thyroid Cancer		
NCT04566848	The Status of Immune Checkpoints at Gastrointestinal Cancer	Completed	Gastrointestinal Cancer	Diagnostic Test: Flow cytometric analysis	
NCT01724320	A Phase I, First-in-man Study of OTX008 Given Subcutaneously as a Single Agent to Patients with Advanced Solid Tumors	Unknown status	Solid Tumors	Drug: OTX008	Phase 1
NCT02575404	Immune Checkpoints in Intraabdominal Ascites Fluid	Active, not recruiting	MelanomaNon-Small Cell Lung CancerSquamous Cell Carcinoma of the Head and Neck	Drug: GR-MD-02Drug: Pembrolizumab	Phase 1
NCT04540159	Validation of Colon Biomarkers for the Early Detection of Colorectal Adenocarcinoma	Recruiting	Colorectal Cancer	Diagnostic Test: Flow-cytometric analysis	
NCT02496260	Safety of GM-CT-01 with and without 5-Fluorouracil in Patients with Solid Tumors	Unknown status	Breast Cancer	Procedure: Research Cardiac MRIProcedure: Biomarkers	
NCT00388700	Pilot Study of Biomarkers and Cardiac MRI as Early Indicators of Cardiac Exposure Following Breast Radiotherapy	Withdrawn	Colorectal Cancer	Drug: GM-CT-01Drug: 5-Fluorouracil, Leukovorin, bevacizumab	Phase 2
NCT00110721	Ex-vivo Evaluation of the Reactivity of the Immune Infiltrate of Cancers to Treatments with Monoclonal Antibodies Targeting the Immunomodulatory Pathways	Terminated	Colorectal Cancer	Drug: GM-CT-01 plus 5-Fluorouracil	Phase 2
NCT01511653	LYT-200 Alone and in Combination with Chemotherapy or Anti-PD-1 in Patients with Metastatic Solid Tumors	Completed	Colon CancerRectal Cancer

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
