# Peer review of "Galectins Are Central Mediators of Immune Escape in Pancreatic Ductal Adenocarcinoma"

_cancers, 2022, doi:10.3390/cancers14225475_

Round 1
Reviewer 1 Report
Main Comment:
This manuscript deals with galectins as mediators of immune escape in pancreatic ductal adenocarcinoma. It provides an overview including illustrative figures. Much work is still to be done to translate these insights into routine clinical practice and this paper may be seen an an incentive to further research in this context.
Additional Comments/Suggestions:
Please use more paragraphs in order to improve the readability (e.g., lines 539-584).
All the abbreviations used in the text should be included in the list of abbreviations ("Appendix A").
Page 3, line 116: over secretion -> oversecretion [or: over-secretion].
Page 4, line 134: …IL-6.IL-6… -> …IL-6. IL-6…; …TAM.PSC… -> …TAM. PSC…
Page 4, lines 138-139: VEGFR2 and NRP-1 form trans-complex, which promotes tumor hypoxia -> VEGFR2 and NRP-1 form trans-complexes, which promote tumor hypoxia [or: VEGFR2 and NRP-1 form a trans-complex, which promotes tumor hypoxia].
Page 4, line 140: …TAM.As…-> …TAM. As…
Page 4, lines 145-146: (This figure was created by ourselves). -> (This figure was created by ourselves.) – The same applies to line 221 on page 6, to lines 272-273 on page 7 and lines 356-357 on page 9.
Page 5, line 166 and line 167: fas -> Fas.
Page 5, line 169: caspase9 -> caspase-9.
Page 5, lines 173-174: This process in which p56lck and ZAP-70 are the most critical. – Please complete this sentence.
Page 5, lines 176-177: The caspase non-dependent pathway-mediated apoptosis under high concentration of galectin-1 [52]. – This sentence is incomplete as well.
Page 6, line 219: t-cell-inhibiting -> T-cell-inhibiting.
Page 7, line 252: HIF-1and -> HIF-1 and.
Page 7, line 262: AMPK-BMI1-Gata2-MICA/B pathway -> AMPK-BMI1-GATA2-MICA/B pathway [in order to be consistent with Fig. 3].
Page 7, line 271: cells.IL-2 -> cells. IL-2.
Page 8, lines 285-288: Galectin-1 recruits and induces PSC and CAF into the extracellular matrix promotes the deposition of associated proteins, remodels the extracellular matrix around PDAC, and ultimately promotes the formation of a fibrotic barrier [76, 77]. – Please clarify this sentence.
Page 10, line 387: lactate dehydrogenase-a -> lactate dehydrogenase A
Page 10, line 399: galectin-3 help pancreatic tumor cells… -> galectin-3 helps pancreatic tumor cells…
Page 11, line 432: Galectin -1 -> Galectin-1.
Page 11, line 448: functions. galecitin-3 -> functions. Galecitin-3.
Page 11, line 458: Galectin-9 help tumor cells… -> Galectin-9 helps tumor cells…
Page 12, lines 470-472: On NK cells, DC cells, the binding of TIM-3 to galectin-9 activates NK cell expression, promotes IFN-gamma secretion and enhances immune function [134]. – Please clarify this sentence.
Page 12, line 481: …use. galectin-9… > …use. Galectin-9…
Page 12, line 488: pathways, (NCT02817633, NCT03099109, NCT02608268)… -> pathways (NCT02817633, NCT03099109, NCT02608268)…
Page 12, line 491: …Dectin-1. combination… -> …Dectin-1. Combination…
Page 12, line 492: …immunosuppression. combined… -> …immunosuppression. Combined…
Page 12 line 494: and enhance anti-tumor immunity. enhance anti-tumor immune function – Please clarify this passage.
Table 1, line 4: Promote… -> Promotes…
Page 13, line 528: galectin1 -> galectin-1.
Page 14, lines 562-564: OTX008 was also able to Michael et al found that the combination of OTX008 and rapamycin inhibited tumor growth much more effectively than rapamycin alone [157]. – Please correct this sentence.
Page 14, lines 567-568: Galectin-3 inhibitor mainly consists of G3-C12, Modified Citrus Pectin (MCP). – Please clarify this sentence.
Page 14, lines 579-580: Another galectin-3 inhibitor: RN1, inhibits… -> Another galectin-3 inhibitor, RN1, inhibits…
Page 14, line 582-584: The current clinical trials of galectins application in solid tumors, which provides a good basis for future clinical application of galectins for the treatment of PDAC, is summarized in Table 2. -> The current clinical trials of galectin application in solid tumors, which provide a good basis for future clinical application of galectins for the treatment of PDAC, are summarized in Table 2.
Page 16, last line: protinase -> proteinase.
Page 17, line 5: Nuclear factor-Like 2 -> Nuclear factor erythroid 2-related factor 2 [consistent with line 111 within the text].
Page 17, line 15: Tumor necrosis factorα -> Tumor necrosis factor α.
Page 17, line 18: Vascular endothlial growth factor -> Vascular endothelial growth factor.
Page 17, line19: Vascular endothlial growth factor receptor2 -> Vascular endothelial growth factor receptor 2.
Appendix A: Throughout this table, upper and lower case should be used in a consistent way.
Page 16, line 610 and line 611: Please correct the punctuation.
Page 16, Line 612: in the writing the manuscript -> in the writing of the manuscript.
Comment concerning the Review Report Form: As this manuscript provides an overview, the questions "Is the work a significant contribution to the field?" and "Is the work scientifically sound?" are not applicable.
Reviewer 2 Report
The manuscript of Tang et al. reviews the involvements of galectins in the proliferation and metastasis of PDAC cells.
In general, the manuscript is well organized and reviews the recent advancements on this topic. However it needs a careful revision to be accepted for publication.
The data are presented in a fragmented way and are often not well connected to each other, making it difficult to read (see for example lines 190-208 of page 5). Often some sentences are repeated several times making the text complicated and confusing (see lines 173,174 page 5, lines 523,524 of page 13,lines 548-550 of page 14).For these reasons I suggest a total revision of the text which also presents some English grammatical and typing errors.
Round 2
Reviewer 2 Report
The authors improved the quality of the manuscript according to all suggestions. The manuscript is suitable for publication.